# Association of Lifelong Intake of Barley Diet with Healthy Aging: Changes in Physical and Cognitive Functions and Intestinal Microbiome in Senescence-Accelerated Mouse-Prone 8 (SAMP8)

**DOI:** 10.3390/nu11081770

**Published:** 2019-08-01

**Authors:** Chikako Shimizu, Yoshihisa Wakita, Makoto Kihara, Naoyuki Kobayashi, Youichi Tsuchiya, Toshitaka Nabeshima

**Affiliations:** 1Frontier Laboratories for Value Creation, SAPPORO HOLDINGS LTD, 10 Okatome, Yaizu, Shizuoka 425-0013, Japan; 2Bioresources Research and Development Department, SAPPORO BREWERIES LTD, 37-1, Nittakizaki, Ota, Gunma 370-0393, Japan; 3Advanced Diagnostic System Research Laboratory, Fujita Health University, 1-98 Dengakugakubo, Kutsukake-cho, Toyoake, Aichi 470-1192, Japan; 4NPO Japanese Drug Organization of Appropriate Use and Research, 3-1509 Omoteyama, Tenpaku-ku, Nagoya, Aichi 468-0069, Japan

**Keywords:** healthy aging, waxy barley, lifelong intake, locomotor activity, spatial recognition, senescence-accelerated mouse prone 8, SAMP8, high-density lipoprotein cholesterol (HDL)

## Abstract

Barley intake reportedly reduces the risk of cardiovascular disease, but effects on the systemic phenotypes during healthy aging have not yet been examined. Therefore, we examined the effects of barley on the lifespan; behavioral phenotypes, such as locomotor activity, and cognitive functions, and intestinal microbiome in the senescence-accelerated mouse-prone 8 (SAMP8) mouse. We prepared two mild high-fat diets by adding lard, in which the starch components of AIN-93G were replaced by rice or barley “Motchiriboshi.” SAMP8 (four weeks old, male) mice were fed AIN-93G until eight weeks old, and then rice (rice group) or barley diet (rice: barley = 1:4, barley group) until death. Changes in aging-related phenotypes, object and spatial recognition, locomotor and balancing activities, and the intestinal microbiome were recorded. Moreover, plasma cholesterol levels were analyzed at 16 weeks old. Barley intake prolonged the lifespan by approximately four weeks, delayed locomotor atrophy, and reduced balancing ability and spatial recognition. Barley intake significantly increased the medium and small particle sizes of high-density lipoprotein (HDL) cholesterol, which is associated with a reduced risk of total stroke. The Bacteroidetes to Firmicutes ratio in the barley group was significantly higher than that in the rice group during aging. Thus, lifelong barley intake may have positive effects on healthy aging.

## 1. Introduction

Barley (*Hordeum vulgare*) is widely consumed as whole grain or pearled barley in a variety of healthy foods, such as bread, cereal, rice, and pasta. It is rich in dietary fiber, and the main component is the mixed linkage of (1, 3) (1, 4)-β-D-glucan similar to oats. The soluble beta-glucan content of barley ranges from 2% to 10% per grain weight depending on the variety, with a generally higher content in waxy barley than in normal varieties [1]. Pearled barley contains much more water-soluble β-glucan than its whole grain, as the β-glucan content in the barley endosperm is higher than that in the bran [2]. The distributional difference of barley β-glucan sets it apart from other grains such as wheat and rice.

Barley has various health benefits, including its ability to reduce blood levels of low-density lipoprotein cholesterol (LDL-Cho), thereby lowering the risk of cardiovascular disease [3,4]; low glycemic index that assists with blood glucose and insulin management [5]; benefits for gut health [6]; and ability to reduce visceral fat [4]. The health benefits of barley have been verified with strong scientific evidence, and health claims have been supported by organizations such as the European Food Safety Authority (EFSA), U.S. Food and Drug Administration (FDA), Health Canada, and Food Standards Australia and New Zealand (FSANZ) [7,8,9]. In Japan, barley rice using pearled barley has historically been consumed as a staple food, but the amount of barley consumption gradually decreased with changing dietary habits. However, as the health benefits of barley have become clearer, the planted area [10] and market size of barley have been increasing yearly.

We previously reported that pearled barley intake reduces blood cholesterol levels and visceral fat in humans over a 12-weeks period [4]. Most published studies of barley’s effects in humans and rodents have been conducted for only several months, which is markedly shorter duration than the average lifespan. However, staple foods like barley are generally consumed as part of habitual diets throughout the lifespan. Many epidemiological studies show a decreased risk of diseases such as coronary heart disease, diabetes, and cancer, as well as mortality with consumption of several different types of whole grains [11,12,13]. However, published studies have mainly focused on lifestyle diseases and mortality, while limited data are available regarding effects on the process of healthy aging, which can be measured by changes in locomotor or cognitive activities. Moreover, epidemiological studies using populations of diverse backgrounds often present conflicting results with difficulty to examine the effects of barley specifically. There are no epidemiological studies on the lifelong intake of barley. Because of the rapid progression of the aging society, promoting healthy aging is one of the most important societal challenges [14]. Therefore, in this study we focus on the impact of lifelong barley intake on aging-related phenotypes.

Many aging studies have considered only a few time points within a certain period, e.g., between young age and old age, and have not been conducted until animal death. On the contrary, for our aging evaluation, we have reported the healthy aging effects of moderate drinking [15] and lemon polyphenols [16] on the lifespan and age-related changes in phenotypes during the lifespan in senescence-accelerated mouse prone 8 or 1 strains established by Takeda et al. [17]. From the perspective of welfare and management of animals, we did not sacrifice the mice but rather examined them using non-invasive pharmacological methods, and their feces to determine the composition of the intestinal microbiome [15,16]. In this study, we prepared mild high-fat diets (fat energy ratio: 27% within recommended human diets) containing barley or rice, and examined not only plasma cholesterol and blood glucose, but also lifespan as well as aging-related behavioral phenotypes such as locomotor activity, balancing power, foot strides, and cognitive functions using SAMP8 mice, which can be used as models of an early cognitive deficit, muscle atrophy and osteoarthritis [18,19,20]. Moreover, we measured plasma LDL-Cho and high-density lipoprotein cholesterol (HDL-Cho) by classifying lipoprotein particle fractions using LipoSEARCH (Skylight Biotech, Akita, Japan), which employs gel permeation high-performance liquid chromatography (HPLC) [21]. This evaluation method was significant for evaluating not only the quantity but also the quality of HDL-Cho and LDL-Cho, which have been reported to affect disease risks [22].

To our knowledge, this is the first report of the effect of barley intake on healthy aging effects during the lifespan of mice.

## 2. Materials and Methods

### 2.1. Ethics Statement

All experiments were approved by the Institutional Animal Care and Use Committee of SAPPORO HOLDINGS LTD. (permit number 2017-003) following the Guidelines for the Proper Conduct of Animal Experiments of the Science Council of Japan.

### 2.2. Diet

The waxy barley “Motchiriboshi” developed by SAPPORO BREWERIES LTD. was pearled by NAGAKURA BARLEY MILLING Co, LTD. (Nagaizumi-cho, Shizuoka, Japan), and then pregelatinized, dried, and powdered by Fuji Shokuhin K. K. (Shizuoka, Japan). The pregelatinized rice was purchased from FRYSTAR CO., LTD. (Yokohama, Japan). We prepared AIN-93G (Standard diet) and two mild high-fat diets (fat content: 27% by adding lard), in which all starch components of AIN-93G were replaced by those of rice or barley as shown in Table 1. The barley or rice, and other ingredients were mixed by Oriental Yeast Co., Ltd. (Tokyo, Japan) to prepare powder diets. Mice were fed AIN-93G and then rice diet for the rice group, only barley diet (approximately for one week), and then a mixture of rice and barley (ratio = 1:4) for the barley group from eight weeks old until death (Figure 1).

The dietary fiber ratios in AIN-93G, the rice and barley diets, as well as a mixture of the rice and barley diets (soluble beta-glucan by HPLC method) were 5% (not measured), 5% (not detected), 7.9% (4.6%), and 7.3% (calculated value: 3.7%), respectively. The dietary fiber composition in the barley diet was entirely derived from the waxy barley, and the soluble and insoluble fiber ratio was 2.5 to 1 as assessed by the modified Prosky method [23]. Other nutritional components such as protein, sugar, and fat were almost equal (Appendix A).

### 2.3. Animals

A total of 36 SAMP8 male mice, four weeks old (Japan SLC, Inc., Hamamatsu, Japan) were acclimated to the animal facility. The floor of the cage was covered with sliced paper, Palmas μ^®^ (Material Research Center, Kawasaki, Japan), which was changed every week. SAMP8 mice with similar body weights were allocated into two groups (rice group: *n* = 18; barley group: *n* = 18) and group housed (three mice per cage; however, three fighting mice in each group were isolated at the age of five weeks before being fed with mild high-fat diets. SAMP8 mice had ad libitum access to tap water and the standard diet, and then the mild high-fat rice diet (rice group) or mild high-fat barley diet (barley group) starting at eight weeks old. Since loose stool was observed in some mice in the barley group, we immediately changed the barley diet to the mixed barley diet (ratio of rice to barley = 1:4, Table 1) until death. The animal facility was maintained at 23 °C ± 1 °C with 55% humidity and a 12-h/12-h light/dark cycle.

### 2.4. Food Consumption, Liquid Consumption, Body Weight, and Survival Analysis

Food consumption and liquid consumption per cage were recorded once or twice per week and three times per week, respectively. These values were expressed as one mouse’s consumption per day, which was calculated by dividing the total consumption per cage by the number of mice in that cage. The body weight of each mouse was examined every week.

### 2.5. Plasma Cholesterol in SAMP8 Mice Consuming Rice or Barley Diets

Foods were removed at 8:30 a.m.–8:40 a.m. Blood sampling from the tail veins of mice started at 14:00 p.m. After centrifugation at 3000 rpm for 5 min, samples were stored at −30 °C until analysis. Plasma cholesterol analysis was conducted in all mice at 16 weeks old.

Plasma lipoprotein was fractionated by HPLC into 20 fractions [21]. The cholesterol concentration in each fraction was measured by LipoSEARCH (Skylight Biotech Inc., Akita, Japan). Okazaki et al. reported that the LDL-Cho lipoprotein subclass, “small and very small size fractions” was significantly higher in patients with coronary artery disease than in normal subjects [24]. Chei et al. reported that small- to medium-sized HDL, but not large HDL, cholesterol levels were inversely associated with total stroke risk [25]. Therefore, we focused on the sum of small and very small sized LDL-Cho particles as “high risk LDL-Cho” and the sum of medium, small, and very small size HDL-Cho as “low risk HDL-Cho,” as well as evaluated the total LDL-Cho and HDL-Cho.

### 2.6. Blood Glucose Analysis

Blood was sampled from tail veins of mice. We directly measured the blood glucose concentration by Precision Xceed G3b smart blue electrodes (Abott, Tokyo, Japan). The limit of glucose detection was 500 mg/dL for blood glucose. Blood glucose analyses were conducted at 16, 29, 42, and 55 weeks old.

### 2.7. Changes in Aging-Related Scores in SAMP8 Mice Consuming Rice or Barley Diets

During the lifespan of all mice, we examined the changes in grading scores, for skin and hair conditions (glossiness, hair coarseness, and hair loss), ulcers, eyes (periophthalmic lesions), and skeleton (spinal curvature) at 9, 11, 16, 21, 27, 31, 36, 41, 46, 51, 54, 59, 64, 68, 72, and 76 weeks old. We evaluated the grading scores as previously reported [26].

### 2.8. CHANGES in Age-Related Physical Activities in SAMP8 Mice Consuming Rice or Barley Diets

Healthy aging aims to maintain function and delay deficits in physical activities such as locomotor activities and balancing abilities. We constantly assessed the locomotor activities and balancing activities by an acryl rod, wire hanging test, and foot print test to determine the starting ages of physical deficits.

#### 2.8.1. Locomotor Activities

Locomotor activity was evaluated for 10 min in boxes (300 × 300 × 350 mm (D × W × H); Brain Science Idea, Inc., Osaka, Japan), of which the bottom was covered with Palmas μ^®^ and quantified using the ANY-maze Video Tracking System (Stoelting Co., Wood Dale, IL, USA). Illumination was provided at 25 lux. A black patch was attached on the back of each mouse to enable tracking in the box with a white-colored floor, as we reported previously [16]. Locomotor activities were measured at 9, 16, 24, 31, 39, 48, 55, 62, 69, and 76 weeks old.

#### 2.8.2. Wire Hanging Test and Balancing Abilities on an Acryl Rod

Balance and prehensile strength were assessed by the wire hanging test [27]. A mouse was put on a metal salamander with a 1-cm square mesh 50 cm above a cushioned surface covered with approximately 3-cm thick Palmas μ^®^. The salamander was inverted and the mouse was attached upside down. The time to fall was measured until 60 s. This trial was repeated three times and used the maximum time. The wire hanging test was conducted at 12, 19, 24, 28, 33, 38, 42, 57, and 62 weeks old.

Tai et al. reported motor incoordination induced by phencyclidine with sodium pentobarbital by measuring the dropping ratio of mice from acryl rods [28]. We modified this method for evaluating the balancing abilities of the SAMP8 mice. A square 5 mm, 320 mm long-acryl rod was bridged on the top edges of a box (300 × 300 × 350 mm (D × W × H)), of which the bottom was covered with approximately 3-cm thick Palmas μ^®^. The mouse was perpendicularly placed on a rod and allowed to balance. The staying ability of mouse was assessed on the rod until 180 s. This trial was repeated three times and used the maximum time. The acryl rod test was conducted at 12, 19, 24, 28, 33, 38, 42, 57, and 62 weeks old.

#### 2.8.3. Foot Print Test

Ko et al. reported that older age was associated with slower self-selected walking speed and shorter stride length in humans [29]. We modified the foot print test previously reported by Blizzard et al. for monitoring foot strides [30]. To prepare apparatus for foot print test, we inserted a reed-shaped white paper into a cardboard cylinder (10 cm wide, 76 cm long). Mice were allowed to walk along a runway of reed-shaped white paper inside a cardboard cylinder (10 cm wide, 76 cm long). All mice were given a few training runs for habituation and one test run after painting four feet with children’s paints (red and blue) (Pentel Co., Ltd., Tokyo, Japan). We measured the “stride, sway, and stance” lengths of both the forefeet and hindfeet of mouse from footprints. Foot prints were measured at 12, 19, 24, 29, 44, 54, and 67 weeks old.

### 2.9. Changes in Object Recognition (Long-Term Object Memory) and Spatial Recognition (Long-Term Location Memory) in SAMP8 Mice Consuming Rice or Barley Diets

We measured the cognitive functions of the novel object recognition test (ORT) and object location test (OLT) as previously reported [15,16,31]. For ORT objects, white golf balls (43 mm diameter) were used as training objects, and a white film case (29 mm (diameter) × 50 mm (height)) was used as the novel object. For OLT objects, green cylindrical wood blocks (44 mm diameter × 44 mm tall put horizontally) were used. Each mouse was free to explore the ORT box without objects for 10 min, once a day for three consecutive days during the ORT habituation phase and for two consecutive days during the OLT habituation phase.

In the ORT and OLT training and test phases, mice were placed in the box and allowed to freely access the two objects for 10 min. The interval between the training and test phases was 24 h for the assessment of long-term memory during the ORT and OLT. The ORT and OLT were performed after more than seven weeks following the previous tests, since the mice forgot the objects used in the previous test, and significantly approached the novel object used seven weeks ago [15]. The ORT experiments were conducted at 10, 17, 25, 32, 40, 49, 56, 63, and 70 weeks old, and the OLT experiments were conducted at 11, 18, 26, 33, 41, 50, 57, and 64 weeks old to evaluate cognitive function. We calculated the recognition index of the objects as previously reported [15,16].

### 2.10. Changes in Intestinal Microbiome in SAMP8 Consuming Rice or Barley Diets

Fresh fecal samples (approximately 100 mg) were collected at the ages of 7, 9, 13, 21, 28, 32, 42, 51, 58, and 67 weeks. The procedure was performed as previously reported [16,32].

### 2.11. Statistical Analyses

All statistical analyses were performed with SPSS software 10.0.7J for Windows (SPSS, Inc., Chicago, IL, USA). Data in the text and figures are presented as the mean ± standard error (SE) of the mean. For locomotor activity, wire hanging test, acryl rod test, foot print test, and aging-related scores, between-group comparisons were performed by two-way analysis of variance (ANOVA) followed by Bonferroni’s post-hoc test for multiple comparisons, if the interactions were significantly different. Between-group comparisons of plasma cholesterol, blood glucose, locomotor activities, ORT, OLT, and bacterial species were performed by unpaired t-tests. Within-group comparisons of blood glucose and foot print test were performed by Dunnett test. In all analyses, a *p*-value <0.05 was considered statistically significant.

## 3. Results

### 3.1. Food Consumption, Liquid Consumption, Body Weight, and Number of Surviving in SAMP8 Mice Consuming Rice or Barley Diets

The results for food consumption, body weight, and number of surviving mice are shown in Figure 2A–C.

There were no significant differences in food consumption between the rice and barley diet groups during standard diet and after changing to the mild high-fat diet, but after 22 weeks, food consumption in the rice group exceeded that in the barley group for some time periods (Table 1, Figure 2A).

There were no significant differences in average body weight between the rice and barley groups until approximately 62 weeks old (Figure 2B). Body weight in both groups increased until 50 weeks old, but the body weight continually decreased only in the barley group at 60–80 weeks old, and then rapidly decreased in both groups. The fluctuating average values of body weight in both groups were due to the deaths of aging mice.

The difference in surviving mice between the groups peaked at 58–60 weeks old, at which time the number of surviving mice in the barley group was higher (Figure 2C). The mean lifespans of the groups were 60.8 ± 5.5 weeks old; and 64.7 ± 4.5 weeks old for the rice and barley groups, respectively (Figure 2C). The average lifespan in the barley group was approximately four weeks longer than that in the rice group.

### 3.2. Plasma Cholesterol Levels at 16 Weeks Old in SAMP8 Consuming Rice or Barley Diets

The plasma LDL-Cho and HDL-Cho levels analyzed by LipoSEARCH^®^ at 16 weeks old are shown in Table 2.

The total LDL-Cho level in the barley group tended to be lower than that in the rice group (*p* = 0.059). The “no high-risk LDL-Cho (large + medium)” in the barley group was significantly lower than that in the rice group (*p* = 0.002), but not the “high-risk LDL-Cho (small + very small)” (*p* = 0.522). Contrastingly, there were no significant differences in total HDL-Cho levels between both the groups (*p* = 0.582). The “low risk HDL-Cho” (medium + small + very small particle size) in the barley group was significantly higher than that in the rice group (*p* = 0.045), but not the “low-risk LDL-Cho (very large and large)” (*p* = 0.871).

### 3.3. Changes in Blood Glucose Levels in SAMP8 Mice Consuming Rice or Barley Diets

We measured the blood glucose at 16, 29, 42, and 55 weeks old (Figure 3A). In the within-group comparison, the blood glucose in both groups significantly increased at 42 weeks old vs. at 16 weeks old (rice group; *p* = 0.004, barley group; *p* = 0.004) but not at 29 weeks old and 55 weeks old by Dunnett test. The blood glucose levels increased with aging, but then decreased in older ages at 55 weeks old. Moreover, the increase in blood glucose levels from 29 weeks old to 42 weeks old in the rice and barley groups was 96.9 ± 12.9 (mg/dL) and 42.7 ± 13.7 (mg/dL), respectively. There was a significant difference in the amount of increase in blood glucose between both groups (Figure 3B, *p* = 0.008). The long-term barley intake appeared to suppress the increase in blood glucose levels.

### 3.4. Changes in Aging-Related Scores in SAMP8 Mice Consuming Rice or Barley Diets

The results for the skin conditions (hair glossiness, and hair loss), periophthalmic lesions, and spinal curvature are shown in Figure 4A–D.

We performed two-way ANOVA to assess the effects of the groups on aging-related scores. Group (F (1,386) = 5.46, *p* = 0.020) and age (F (15,386) = 46.2, *p* < 0.001) were significantly associated with the scores of periophthalmic lesions (Figure 4A). There was no significant interaction between group and age (F (15,386) = 0.41, *p* = 0.975). In hair glossiness, group (F (1,386) = 89.3, *p* < 0.001) and age (F (15,386) = 47.3, *p* < 0.001) were significantly associated with the scores (Figure 4B). There was a significant interaction between group and age (F (15,386) = 2.29, *p* = 0.004). Using the Bonferroni’s post-hoc test, the scores of hair glossiness in the barley group were significantly lower than those in the rice group after the age of 27 weeks (*p* < 0.001; 27, 36, 54 weeks old, *p* = 0.002; 31 weeks old, *p* = 0.003; 41 and 64 weeks old, *p* = 0.008; 46 weeks old, *p* = 0.016, 51 weeks old, *p* = 0.001; 68 weeks old, *p* = 0.004; 72 weeks old), but not at 59 and 76 weeks (*p* > 0.05). In terms of hair loss, group (F (1,386) = 84.2, *p* < 0.001) and age (F (15,386) = 2.27, *p* = 0.004) were significantly associated with the scores (Figure 4C). There was a significant interaction between group and age (F (15,386) = 2.24, *p* = 0.005). Using the Bonferroni’s post-hoc test, the scores of hair loss in the barley group were significantly lower than those in the rice group after the age of 16 weeks (*p* = 0.006; 16 weeks old, *p* = 0.038; 21 weeks old, *p* = 0.024; 36 weeks old, *p* = 0.043; 41 weeks old, *p* = 0.028; 46 weeks old, *p* = 0.022; 51 weeks old, *p* = 0.024; 54 weeks old, *p* = 0.011; 68 weeks old, *p* < 0.001; 72, and 76 weeks old). In terms of spinal curvature age (F (15,386) = 22.3, *p* < 0.001), but not group (F (1,386) = 3.57, *p* = 0.059), was significantly associated with the scores (Figure 4D). There was no significant interaction between group and age (F (15,386) = 0.64, *p* = 0.840).

The long-term intake of barley resulted in the decline in hair conditions such as reduced hair glossiness and promoted hair loss.

### 3.5. Changes in Age-Related Physical Activities in SAMP8 Mice Consuming Rice or Barley Diets

To ensure healthy aging, it is necessary to maintain physical activities such as locomotor activity, muscle strength, and balancing ability. We investigated whether lifelong barley intake protects against age-related impairment of motor activities.

#### 3.5.1. Locomotor Activities

We examined the changes in locomotor activity from 9 to 76 weeks old (Figure 5). We performed two-way ANOVA to assess the effects of the group on locomotor activities. Group (F (1,229) = 17.4, *p* < 0.001) and age (F (15,229) = 41.6, *p* < 0.001) were significantly associated with the locomotor activities (Figure 5). Locomotor activities in the barley group were significantly higher than those in the rice group. Although there was no significant interaction between group and age (F (15,229) = 0.43, *p* = 0.917), we conducted the between-group comparison at each age. There was a significant difference between the rice and barley groups at 39, 48, and 55 weeks old (*p* = 0.015, 0.028, and 0.023, respectively). From the results, the long-term intake of barley delayed the locomotor atrophy induced by aging as compared with the rice diet.

#### 3.5.2. Wire Hanging Test and Balancing Ability on an Acryl Rod

We examined the changes in muscle coordination and endurance by wire hanging from 12 to 62 weeks old (Figure 6A).

We performed two-way ANOVA to assess the effects of the group on muscle coordination. Age (F (9,277) = 15.1, *p* < 0.001), but not group (F (1,277) = 1.31, *p* = 0.253) was significantly associated with the muscle coordination (Figure 6A). There was no significant interaction between group and age (F (9,277) = 1.80, *p* = 0.068). There was no significant difference in time to fall between the rice and barley groups, although significant decreases with age were observed in muscle coordination (*p* < 0.001, Figure 6A).

Next, we examined the changes in balancing abilities on an acryl rod from 12 to 62 weeks old (Figure 6B). Prior to approximately 30 weeks old, many of the SAMP8 mice had a habit of jumping off the rod toward the bottom, and this method seemed not to be suitable (inside broken line in Figure 6B), but after 30 weeks of age they stopped this habit, and we could measure the abiding time on a rod. We performed two-way ANOVA followed by Bonferroni’s post-hoc test to assess the effects of the group on balancing abilities. Group (F (1,277) = 21.1, *p* < 0.001) and age (F (9,277) = 12.2, *p* < 0.001) were significantly associated with the balancing abilities (Figure 6B). There was a significant interaction between group and age (F (9,277) = 2.87, *p* = 0.003). Using the Bonferroni’s post-hoc test, the time to fall in the barley group was significantly prolonged compared to that in the rice group at 42 weeks old (*p* < 0.001), 57 weeks old (*p* = 0.042), and 62 weeks old (*p* < 0.001).

#### 3.5.3. Foot Print Test

The measuring points of the foot print test were six points per mouse (stride, sway, and stance × forefoot and hindfoot, Figure 7A) [30]. The results of the foot print test from 12 to 67 weeks old are shown in Figure 7B–D (stride, sway, and stance of forefoot, respectively), and Figure 7E–G (stride, sway, and stance of hindfoot, respectively).

We performed two-way ANOVA to assess the effects of the group on six distances of foot point. Age (F (6,190) = 12.6, *p* < 0.001), but not group (F (1,190) = 0.88, *p* = 0.348) was significantly associated with the stride of forefoot (Figure 7B). There was no significant interaction between group and age (F (6,190) = 1.38, *p* = 0.223). Age (F (6,190) = 12.1, *p* < 0.001), but not group (F (1,190) = 0.013 *p* = 0.910) was significantly associated with the sway of forefoot (Figure 7C). There was no significant interaction between group and age (F (6,190) = 1.00, *p* = 0.425). Age (F (6,190) = 5.51, *p* < 0.001), but not group (F (1,190) = 2.47, *p* = 0.118) was significantly associated with the stance of forefoot (Figure 7D). There was no significant interaction between group and age (F (6,190) = 0.45, *p* = 0.845). Age (F (6,190) = 12.6, *p* < 0.001), but not group (F (1,190) = 0.03 *p* = 0.862) was significantly associated with the stride of hindfoot (Figure 7E). There was no significant interaction between group and age (F (6,190) = 0.71, *p* = 0.643). Age (F (6,190) = 15.7, *p* < 0.001), but not group (F (1,190) = 1.24 *p* = 0.267) was significantly associated with the sway of hindfoot (Figure 7F). There was no significant interaction between group and age (F (6,190) = 1.22, *p* = 0.298). Only in the stance of hindfoot, group (F (1,190) = 5.14, *p* = 0.025) and age (F (6,190) = 16.3, *p* < 0.001) were significantly associated with the distances (Figure 7G). There was a significant interaction between group and age (F (6,190) = 0.23, *p* = 0.968). We conducted the between-group comparison by unpaired t-test at each age. Only the sway (hind foot) at 12 weeks old, and the stance (hind foot) at 24 weeks old in the barley group were significantly larger than those in the rice group (the circle surrounding the symbols in Figure 7F,G, *p* = 0.032 and *p* = 0.031, respectively).

In within-group comparison vs. 12 weeks old (the youngest age of this measurement) by Dunnett test, the significant decrease of foot prints (stride, sway of forefoot, and sway of hindfoot) was observed at 19 weeks old (Figure 7B,C,F).

### 3.6. Changes in Object Recognition (Long-Term Object Memory) and Spatial Recognition (Long-Term Location Memory) in SAMP8 Mice Consuming Rice or Barley Diets

Maintaining cognitive functions as well as physical abilities is essential for healthy aging. The recognition indices of familiar and novel objects during the test phases of ORT in the rice and barley groups are illustrated in Figure 8A,B (ORT), and Figure 8C,D (OLT), respectively. The circles surrounding the symbols of familiar and novel (location) objects recognition indexes indicate a significant difference during the test phase, but no significant difference during the training phase (Figure 8A–D).

While no significant differences were observed in the ORT during the training phase, in the test phase conducted from 10 to 70 weeks old, the recognition index for the novel object (film case) was significantly higher than for the familiar object (golf ball) in both the rice and barley groups (*p* < 0.05), but not in the rice group at 49 weeks old, and in both group at 70 weeks old (Figure 8A,B). From the results, we concluded that there were no significant differences in object recognition ability between both groups throughout life.

No significant differences were observed in the OLT during the training phase regarding the recognition index of familiar and novel locations, while during the test phase from 11 to 65 weeks old the mice recognized the novel position significantly more than the familiar location only at 11 and 57 weeks old in the rice group (*p* < 0.05; Figure 8C), and from 11 to 41 and 57 weeks old in the barley group (*p* < 0.05; Figure 8D). The disorder of spatial recognition in the OLT (Figure 8C,D) preceded the impairment of object recognition in the ORT (Figure 8A,B), as we reported previously [15,16].

### 3.7. Changes in Intestinal Microbiome in SAMP8 Consuming Rice or Barley Diets

UniFrac analysis is an effective distance metric for microbial species [33] that visually expresses the composition of bacterial species at a specific site. Initially, the overall structure of the intestinal microbiome was evaluated by unweighted UniFrac analysis using all data at seven weeks old (fed standard diet), and 9, 13, 21, 28, 32, 42, 51, 58, and 67 weeks old (fed mild high-fat rice or barley diets) (Figure 9A).

The intestinal microbiome of both groups at seven weeks old (fed standard diet) was the same; however, differences were observed between the rice and barley groups from 9 to 67 weeks old. Moreover, we did not distinguish colors among ages in Figure 9A, but the microbe compositions in both groups shifted in parallel in the direction of the arrows (Figure 9A) from 9 to 67 weeks old. The intestinal microbiome was largely changed by diets and ages.

Subsequently, the microbiome composition at the phylum level was evaluated. In between-group comparisons of the rice group with the barley group, the level of Bacteroidetes/Firmicutes in the barley group was significantly higher than that in the rice group (Figure 9B) at 9 (*p* = 0.019), 13 (*p* < 0.001), 21 (*p* = 0.012), and 32 weeks old (*p* = 0.025), although the data at nine weeks old were obtained just before the barley diet was changed to the 20% rice diet-mixed barley diet. The higher levels of Bacteroidetes/Firmicutes in the barley group than in the rice group continued until 64 weeks old, although there was no significant difference after 42 weeks old.

Genus-level differences in the microbiome between both groups were evaluated. The level of *Bacteroides* (phylum Bacteroidetes) in the barley group was significantly higher than that in the rice group (*p* = 0.002) at nine weeks old, but after then decreased (*p* > 0.05, Figure 9C). Furthermore, the level of *Prevotellaceae;g*_ (phylum Bacteroidetes) in the barley group was significantly higher than that in the rice group at 13, 21, 28, 32 (*p* < 0.001), and 51 weeks old (*p* = 0.012, Figure 9D).

## 4. Discussion

The food intake in the rice group appeared to exceed that in barley group during several time periods (Figure 2A). Three mice in the rice group and one mouse in the barley group suffered severe diabetes indicated by excessive urination and a strong positive reaction in urine glucose test (data not shown), which could lead to greater food intake. Therefore, the average food intake values could be affected in these four diabetic mice. We speculated that the difference in the rate of severe diabetes indicates prevention of diabetes by barley intake, as three mice in the rice group and one in the barley group suffered from diabetes.

There were no significant differences in average body weight between the rice and barley groups until 50 weeks old, but the body weight decreased only in the barley group after 60 weeks old (Figure 2B). The fluctuating average values of body weight in both groups were caused by weight loss before death of aging mice. In our previous study using SAMP8 mice fed a pellet diet (CRF-1, 13% fat energy ratio, Charles River Laboratories, Yokohama, Japan) and tap water [15], the average body weight was approximately 30 g and peaked at 30 weeks old. Body weight of mice in the previous study peaked at a younger age than that of mice in this study, which were fed mild high-fat diet and had a body weight of more than 45 g, with a peak at 50 weeks old (Figure 2B). Body weight of mice in this study was 1.5-fold higher and the maximum peak of body weight occurred at approximately 20 weeks later (Figure 2B) compared to that of mice in our previous study [15]. Yamamoto et al. reported that fat absorption ability decreases with age due to the degeneration of villi in the small intestine, and an oral high-fat diet (fat energy ratio; 30%) inhibited the attenuation of lipid absorption ability in SAMP8 mice [34]. We speculate that the mild high-fat diets (energy ratio; 27%) may also prevent the attenuation of lipid absorption ability with aging in this study. Further, Miyamoto et al. reported that the weight gain of mice fed a high-fat diet (energy ratio; 60%) containing high levels of barley fiber was lower than that of control mice fed a diet with 5% cellulose [35], but no significant differences between the rice and barley groups were observed in our study (Figure 2B). We suggest that the constitutional difference between SAMP8 and C57BL/6J mice in glucose tolerance [36] might lead to differences in weight gain, as the skyrocketing increase of body weights in the rice and barley groups until 30 weeks old was far superior to the barley–fiber function in weight gain. Over 60 weeks of age, the reduction of body weight in mice fed with the barley diet was remarkable (Figure 2B). The results can be explained by the attenuation of nutrient absorption by intestinal aging, as the barley fiber intake reduces blood glucose elevation [37]. The obesity induced by high-fat diets leads to an increase in lifestyle-related diseases and remains a major global and societal challenge [38]. On the contrary, Matsuo et al. reported that the risk nadir BMIs (BMIs with the lowest mortality) for men in the age groups of 40–59 and 60–79 years were 23.4 and 25.3 kg/m^2,^ respectively in Japanese [39]; underweight (BMI < 20) individuals aged 65–79 years have an increased risk of all-cause mortality than those with a BMI between 20.0 and 29.9 in the Japanese population [40]. We should rethink the age-appropriate nutrition and body weight goals for individuals before middle age, and those who are elderly.

The difference in the number of surviving mice between both groups peaked at 58–60 weeks old (Figure 2C). On the contrary, the mean lifespan of mice fed with a low-fat diet (CRF-1) [15] was approximately six weeks longer (71.0 ± 5.8 weeks) than that in the barley group in the present study. Our findings showed that the lifelong intake of a diet rich in barley fiber and mildly high in fat has clear positive effects on lifespan, but these effects were not as high as those of low-fat diet. For a healthy diet, we should pay attention to the fat content of food, especially during early and middle age. Moreover, the habit to eat barley prior to the start of significant aging may help prolong the lifespan.

The health claims on food labels in the European Union and the United States that barley consumption can contribute to lowering blood cholesterol and reducing the risk of coronary heart disease, mainly cite barley’s role in reducing LDL-Cho [3,4]. In our study, the total LDL-Cho in the barley group also tended to be lower than that in the rice group (Table 2). Moreover, the “no high-risk LDL-Cho (large + medium)” in the barley group was significantly lower than that in the rice group, but not the “high-risk LDL-Cho (small + very small),” which was related to cardiac infarct [24]. On the contrary, three mice in the rice group and one mouse in the barley group (four SAMP8 mice total) showed severe diabetes as demonstrated by excessive urination and strong positive urine glucose tests as mentioned above (data not shown). The average “high-risk LDL-Cho” concentration in the four diabetic mice at 16 weeks old, was 32.8 ± 3.98 mg/dL and was significantly higher than that in the non-diabetic mice (14.3 ± 1.04 mg/dL, *p* < 0.001). Therefore, the “high risk LDL-Cho” measure may be a diagnostic index of pathogenesis of progressing diabetes.

On the contrary, previous studies failed to find significant effects of barley on total HDL-Cho [4,41]. In our study, there were also no significant differences in total HDL-Cho between both groups; however, barley intake significantly increased the “low risk HDL-Cho” (medium + small + very small particle size), which correlates with a reduced risk of stroke [25]. This result indicated the importance of HDL-Cho quality, and not just the quantity of total HDL-Cho. Based on our findings, barley intake increased the low risk, good quality HDL-Cho as well as reducing total LDL-Cho and no-high risk LDL-Cho, either off or on, which are related to the risk of circulatory diseases [42]. This is a new aspect of barley effects on HDL-Cho.

The SAMP8 mice exhibit impaired glucose tolerance [43]. In terms of the short-term effects of barley on blood glucose, barley is considered one of the low glycemic index foods due to its β-glucan content [5], which results in the reduction of postprandial elevation of blood glucose levels. Tosh et al. reported that the intact grains as well as a variety of processed oat and barley foods can significantly reduce post-prandial blood glucose levels [44]. In our study, the long-term intake of barley also prevented increasing levels of blood glucose (Figure 3B), which indicated a reduction in diabetes risk. The increase in blood sugar level was similar (*p* = 0.007) even in the non-diabetic mice (blood sugar levels > 500 mg/dL). Schulze et al. reported that higher intake of cereal fiber and magnesium, but not vegetable or fruit fiber, decreases diabetes risk [45]. The long-term intake of barley rather than rice delayed the development of pathogenic diabetes.

Mehla et al. reported that a high-fat diet (fat energy ratio: 60%) induced type 2 diabetes in SAMP8 mice [36]. In our study while feeding the mild high-fat diet (fat energy ratio 27%), only four mice suffered severe diabetes, and these mice showed more rapid increases in body weight than the other mice. Three of the four diabetic mice were bred in individual cages, leading to increased food consumption. Moreover, the mean blood glucose level at 55 weeks old decreased, except for one surviving mouse with severe diabetes (whose blood glucose was more than 500 mg/dL). We assumed that decreased nutrient absorption due to aging [34] resulted in the low levels of blood glucose (reduced approximately 60 mg/dL) in SAMP8 mice. Because the susceptibility of SAMP8 mice to diabetes was affected by the fat–energy ratio (low or high fat diet), the ability of intestinal absorption, and breeding circumstance (group-breeding or not), the SAMP8 mice might be a suitable type 2 diabetes model for humans, and not only a model of dementia or locomotor defect with aging.

Indicators of aging were monitored starting at 16 weeks old in mice fed with a mild high-fat diet, and the long-term intake of barley resulted in a delay of aging of hair conditions such as reduced hair glossiness and hair loss compared with a rice diet (Figure 4B,C). The grading score and incidence of SAMP8 mice began to increase from four or six months of age and continued to increase with advancing age [26]. Oxidative stress in SAMP8 mice was higher than that in SAMR1 mice (senescence resistant mouse) [46]. Diabetes promotes oxidative stress associated with metabolic diseases [47]. The hyperglycemia in diabetes causes protein glycation and advanced glycation end products (AGEs) formation that lead to oxidative stress, release of cytokines, and establishment of secondary complications such as neuropathy, retinopathy, and nephropathy [48]. Matilainen et al. reported that early androgenetic alopecia could be a clinical marker of insulin resistance [49]. We considered that the higher blood glucose in the rice group likely promoted the progress of hair aging compared with that in the barley group (Figure 4B,C), although the hair loss in SAMP8 mice may not be equivalent to androgenetic alopecia.

In age-related physical activities [50], the locomotor activities (Figure 5) and the muscle coordination abilities in the wire-hanging test (Figure 6A) significantly decreased with age. The obesity-induced oxidative stress promoted by diabetes accelerates functional decline with age [51,52]. Guo et al. revealed that gastrocnemius muscle mass peaked at seven months and functional and structural decline was observed at eight months in SAMP8 [53]. Moreover, SAMP8 mice develop muscle atrophy induced by a high-fat diet which correlates with insulin resistance [19]. Further, SAMP8 mice progress to osteoarthritis and histological joint degeneration with aging [20]. We consider that the locomotor defects by aging (Figure 5) were caused by the innate character of SAMP8 mice and promoted by a mild high-fat diet that induces oxidative stress. In the between-group comparisons, the long-term intake of barley significantly retarded the progression of locomotor disorders compared with rice (Figure 5). Moreover, the balancing abilities assessed with an acryl rod were superior in mice fed the barley diet as compared to that in mice fed the rice diet (Figure 6B). Wu et al. reported an association between dietary fiber intake and physical performance in older adults [54]. Oat bran and barley fiber intake reduces the oxidative stress induced by a high-fat diet in pigs [55]. Glycemic control is associated with maintenance of lower-extremity function in older adults with diabetes [56]. We assume that long-term intake of barley containing high levels of dietary fiber helps to prevent locomotor and balancing disorders through reducing oxidative stress and inflammation. On the contrary, there was no significant difference between the barley and rice groups in the wire hanging test for muscle coordination (Figure 6A), which did not correlate with locomotor activities (Figure 5).

The foot print test can detect gait abnormality. Schwenk et al. reported that the stride length in humans was significantly decreased between non-frail and prefrail [57]. In our study, the six points of foot prints (stride, sway, and stance in forefoot and hindfoot), as well as locomotor activities, were monitored to assess gait impediments with age (Figure 7B–G). In between-group comparisons, the distance s of foot prints at older ages were not significantly different. The SAMP8 mice develop not only muscle loss but also osteoarthritis with age [20]. The wire-hanging movement (muscle coordination) decreased much more rapidly than locomotor activities and foot print movement (the flat movement).

The AGEs formation, which leads to the oxidative stress and inflammation, gradually progresses with advanced age without diabetes. Momma et al. reported that participants with higher skin auto fluorescence (a noninvasive method for measuring tissue AGEs), had lower muscle strength and power [58]. The barley has a low glycemic index value that assists with blood glucose management [5]. We assume that the long-term intake of barley might contribute in delaying the AGEs formation with age in healthy humans as well as diabetic humans.

SAMP8 mice are a neuropathological model of accelerated brain aging and dementia [17,18]. Palomera-Ávalos reported that the high-fat diet induced cognitive disorder with increase of inflammatory parameters in aged mice [59]. Moreover, diabetes reportedly increases the risk of both Alzheimer’s disease and vascular dementia [60]. High blood glucose levels and insulin resistance induce increased oxidative stress as well as reduced degradation of amyloid beta, which is closely associated with dementia [61]. In our study, the results of the OLT showed that location-related memory in the barley group was better than that in the rice group (Figure 8C,D). The disorder of spatial recognition in the OLT preceded the impairment of object recognition in the ORT, as we reported previously [15]. In a recent study [35], Miyamoto et al. reported that a high barley beta-glucan diet increases glucagon like peptide-1 (GLP-1) and peptide YY secretion by improving insulin sensitivity. Moreover, diabetes therapeutic medicines such as pioglitazone improve insulin tolerance, and GLP-1 improves diabetes [62,63]. Hansen et al. reported that the GLP-1 receptor agonist liraglutide improves memory function in SAMP8 mice [64]. In this study, the barley diet suppressed the increase of blood glucose (Figure 3A,B) and might presumably promote the release of GLP-1, leading to a delayed failure of spatial recognition induced by age and a mild high-fat diet.

Recently, a relationship has been demonstrated between the gut microbiome and dementia in humans [65]. In our study, UniFrac analysis indicated that the intestinal microbiome of both groups at seven weeks old (fed standard diet) was the same, as expected, after which differences were developed between the rice and barley groups from 9 to 67 weeks old (Figure 9A). Moreover, the microbe compositions in both groups shifted in the direction of the white arrows “aging” (Figure 9A) from 9 to 67 weeks old. Our results confirmed the data of previous reports that the intestinal microbiome changes by age [66] and by barley intake [67]. Moreover, the changes in microbiome composition in both groups shifted in parallel with age (Figure 9A). We previously reported that lemon polyphenols with anti-oxidant abilities prevented the aging of the intestinal microbiome as shown by UniFrac analysis [16]. Changes in the intestinal microbiome occurred rapidly and remarkably after barley intake at nine weeks old (Figure 9A). However, the inhibition of microbiome changes with age by barley seemed to be weaker than that by lemon polyphenols (LPP) with high radical scavenging power [16], although barley contains bioactive phytochemicals besides beta-glucan [68].

Subsequently, the microbiome composition at the phylum level was evaluated (Figure 9B). In between-group comparisons of the rice group with the barley group, the level of Bacteroidetes/Firmicutes in the barley group was shown to be significantly higher (Figure 9B) at 9, 13, 21, and 32 weeks old, although the data at nine weeks old were obtained just before the barley diet was changed to a 20% rice diet-mixed barley diet. Huazano-García reported that high-fat diet feeding for five weeks decreased the level of Bacteroidetes/Firmicutes associated with body gain [69]. A high-fat diet containing dietary fiber (a mixture of soluble fiber and insoluble fiber) for 16 weeks reportedly yielded a higher level of Bacteroidetes/Firmicutes than a diet containing only soluble or insoluble fibers [70]. The barley “Motchiriboshi” that we used, contains both soluble fiber and insoluble fiber (at a ratio of 2.5:1). The barley fiber effectively increases the Bacteroidetes/Frmicutes ratio, and maintained the higher ratio as compared with cellulose in the rice diet with aging in our study.

Genus-level differences in the microbiome between both groups were evaluated (Figure 9C,D). The level of *Bacteroides* (phylum Bacteroidetes) in the barley group was significantly higher than that in the rice group at nine weeks old. Furthermore, the level of *Prevotellaceae;g_* (phylum Bacteroidetes) in the barley group was significantly higher than that in the rice group. Wang et al. reported that intake of high molecular weight barley beta-glucan for five weeks altered the microbiome profile of groups such as *Bacteroides* and *Prevotella*, which were correlated with shifts in risk factors of cardiovascular disease, including body mass index, waist circumference, blood pressure, and triglyceride levels [67]. Moreover, Kovatcheva-Datchary et al. reported that, three days of consumption of barley kernel-based bread (short-term intake) increases the *Prevotella*/*Bacteroides* ratio, and *Prevotella* is proven to play a role in glucose metabolism by promoting increased glycogen storage in germ-free mice [71]. We suggest that the increase in *Prevotellaceae;g_* by barley intake (Figure 9D) also contributed to a sustained improvement in glucose metabolism (Figure 3A,B) as well as the low glycemic response. Our results suggest that lifelong intake of barley has healthy aging effects not only on overall host health but also on the intestinal environment.

In most previous reports, the test periods in rodents and human have been less than several months. Our findings of the effects of a barley diet on the intestinal microbiome corroborate those of previous reports by short-term intake of dietary fiber [67,70,71], but newly confirmed the lifelong effect of barley diet (Figure 9A–D).

The short-term effects of foods may not be the same as the lifelong effects. Therefore, we should evaluate effects of foods on healthy aging to understand the effects of lifelong intake to avoid misleading conclusions.

## 5. Conclusions

Barley intake in SAMP8 mice significantly increased low-risk HDL-Cho levels, and reduced LDL-Cho. Barley intake prolonged the lifespan by approximately four weeks. Moreover, long-term barley intake delayed aging-related phenotypes, locomotor atrophy, loss of balancing abilities, and reduced spatial memory. The Bacteroidetes to Firmicutes ratio in the barley group was significantly higher than that in the rice group, and decreased with aging, but the barley group maintained a higher ratio. In terms of genus-level differences in the microbiome, the levels of *Bacteroides* (phylum Bacteroidetes) and *Prevotellaceae;g_*(phylum Bacteroidetes) in the barley group were significantly higher than those in the rice group. Thus, lifelong barley intake, i.e., habit of consuming barley containing high levels of dietary fiber may have positive effects on both phenotypes and the intestinal environment during the healthy aging process.

## Figures and Tables

**Figure 1 nutrients-11-01770-f001:**
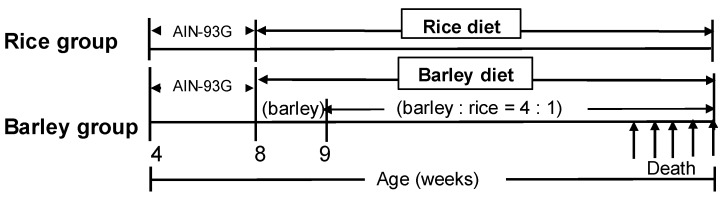
Protocol of feeding.

**Figure 2 nutrients-11-01770-f002:**
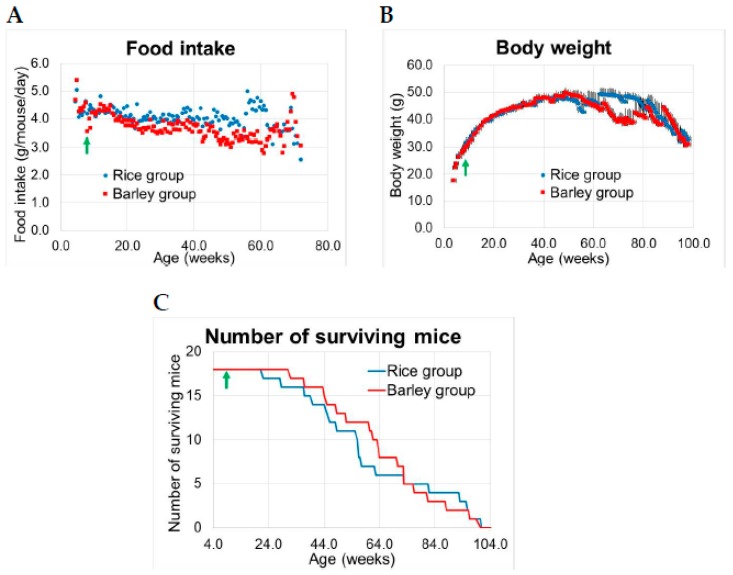
Food consumption, body weight, and number of surviving senescence-accelerated mouse-prone 8 (SAMP8) mice consuming rice or barley diets during the lifespan: (**A**) food consumption (g/mouse/day), (**B**) body weight (g), mean ± SE, (**C**) number of surviving mice. Green arrow; start timing of mild high-fat diets.

**Figure 3 nutrients-11-01770-f003:**
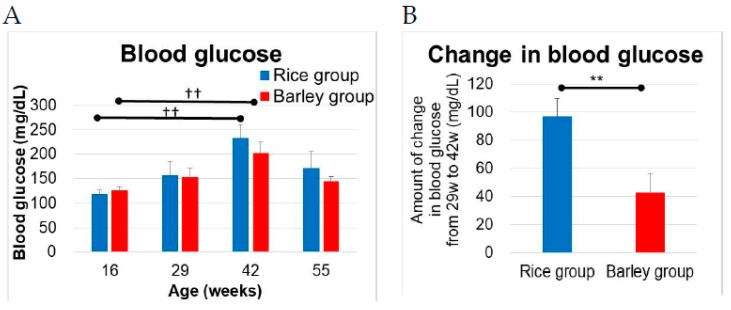
Changes in blood glucose in SAMP8 mice consuming rice or barley diets: (**A**) Levels of blood glucose at 16, 29, 42, and 55 weeks old; (**B**) amount of change in blood glucose from 29 to 42 weeks old. The blood glucose in both groups increased with age from 16 to 42 weeks old. The blood glucose levels of one mouse in the rice group and one mouse in the barley group of age 42 weeks, as well as one mouse in the rice group of age 55 weeks were over the limit of detection and were therefore set at 500 mg/dL. ††; *p* < 0.01; within group comparison at 29, 42, 52 weeks old vs. at 16 weeks old by Dunnett test. ** *p* < 0.01; rice group vs. barley group by unpaired *t*-test.

**Figure 4 nutrients-11-01770-f004:**
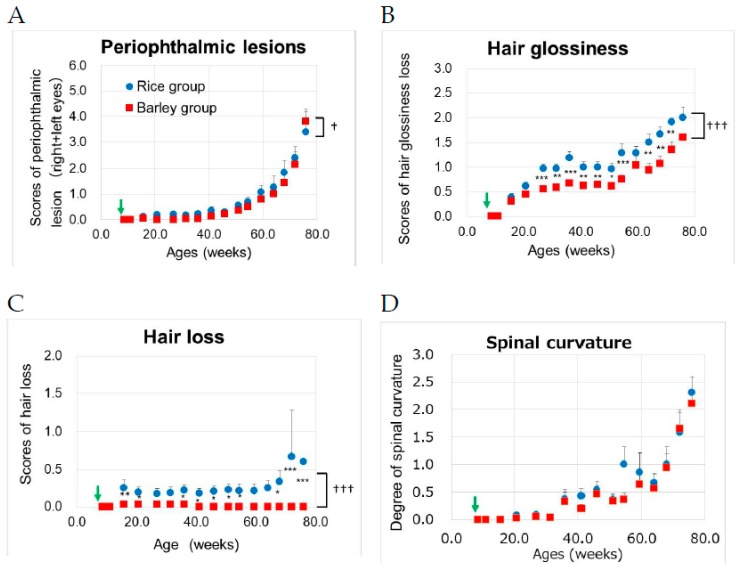
Changes in aging-related scores [26] in SAMP8 mice consuming rice or barley diets during the lifespan: (**A**) periophthalmic lesions; (**B**) hair glossiness; (**C**) hair loss; (**D**) spinal curvature. By two-way ANOVA, ††† *p* < 0.001, † *p* < 0.05; group, * *p* < 0.05, ** *p* < 0.01, *** *p* < 0.001; rice group vs. barley group by the Bonferroni’s post-hoc test for multiple comparison. Green arrow; start timing of mild high-fat diets.

**Figure 5 nutrients-11-01770-f005:**
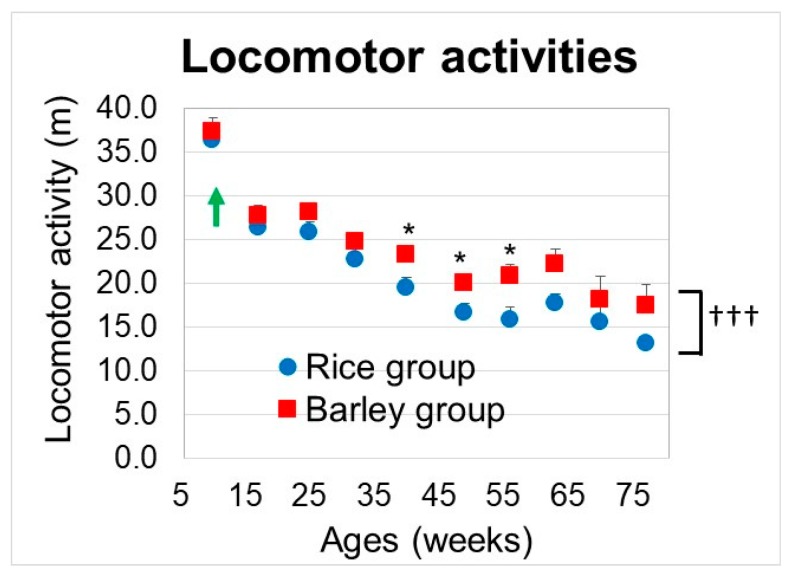
Changes in locomotor activities in SAMP8 mice consuming rice or barley diets during the lifespan. By two-way ANOVA, ††† *p* < 0.001; group, * *p* < 0.05; between-group comparison at 39, 48, and 55 weeks old by unpaired *t*-test. Green arrow; start timing of mild high-fat diets.

**Figure 6 nutrients-11-01770-f006:**
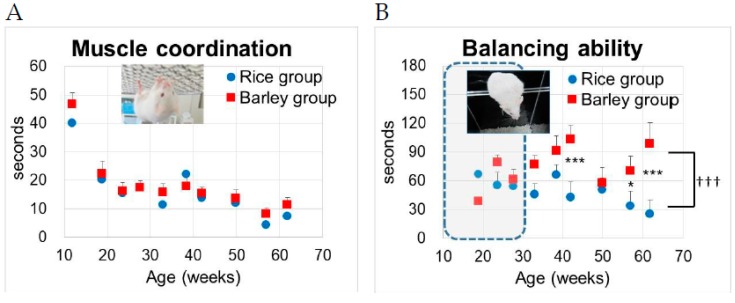
Changes in muscle coordination and endurance balancing in SAMP8 mice consuming rice or barley diets during the lifespan: (**A**) wire hanging test; (**B**) balancing on an acryl rod. By two-way ANOVA, ††† *p* < 0.001; -group, *** *p* < 0.001, * *p* < 0.05 by Bonferroni’s post-hoc test for multiple comparisons.

**Figure 7 nutrients-11-01770-f007:**
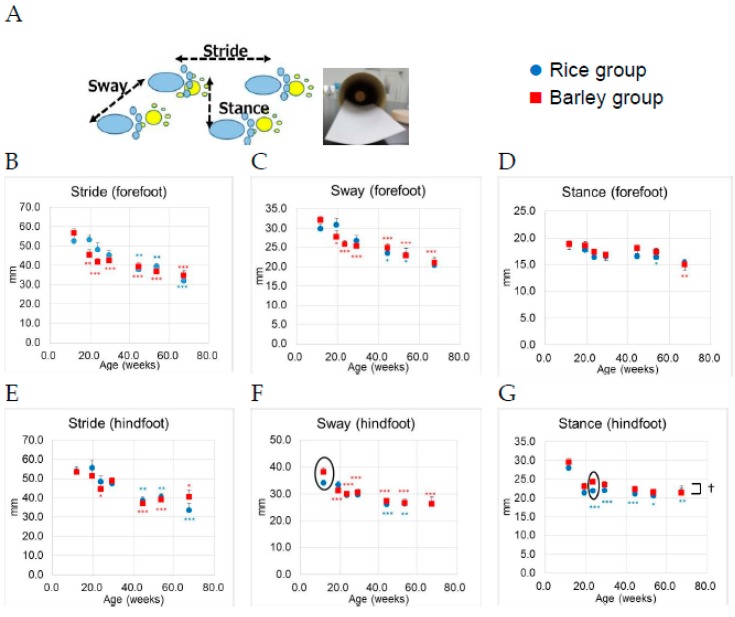
Changes of parameters in the foot print test in SAMP8 mice consuming rice or barley diets during the lifespan: (**A**) Measurement of six points in the foot print test (sway, stride, and stance in forefoot and hindfoot); (**B**) stride of forefoot; (**C**) sway of forefoot; (**D**) stance of forefoot; (**E**) stride of hindfoot; (**F**) sway of hindfoot; (**G**) stance of hindfoot. By two-way ANOVA, † *p* < 0.05; group. The circle surrounding the symbols indicate a significant difference between groups by unpaired *t*-test (*p* < 0.05). * *p* < 0.05; ** *p* < 0.01; *** *p* < 0.01 (blue color; Rice group, red color; Barley group); within group comparison by Dunnett test vs. 12 weeks old.

**Figure 8 nutrients-11-01770-f008:**
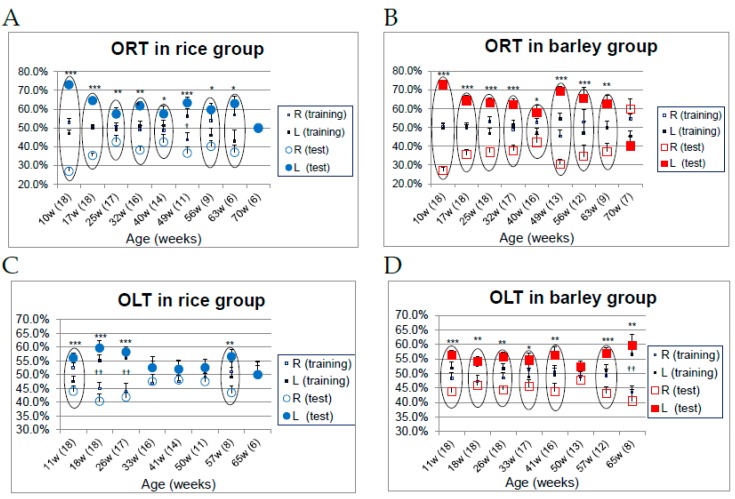
Changes in object recognition (long-term object memory) and spatial recognition (long-term location memory) in SAMP8 mice consuming rice or barley diets during the lifespan: (**A**) Recognition indexes of ORT in the rice group; (**B**) recognition indexes of ORT in the barley group; (**C**) recognition indexes of OLT in the rice group; (**D**) recognition indexes of OLT in the barley group. ORT; novel object recognition test, OLT; object location test. The numbers in parentheses show the number of surviving mice at the different ages. †† *p* < 0.01, † *p* < 0.05; comparison of recognition index of the right object vs. the left object in training phase by unpaired t-test. *** *p* < 0.001, ** *p* < 0.01, * *p* < 0.05; comparison of recognition index of the familiar object (location) vs. the novel object (location) in test phase by unpaired *t*-test.

**Figure 9 nutrients-11-01770-f009:**
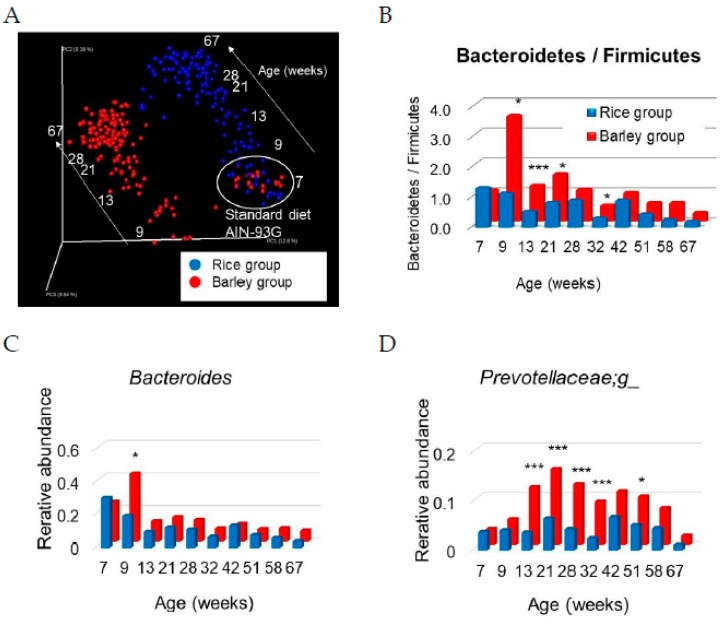
Changes in the intestinal microbiome in SAMP8 mice consuming rice or barley diets: (**A**) UniFrac analyses (unweighted); (**B**) Bacteroidetes (phylum)/Firmicutes (phylum); (**C**) *Bacteroides* (genus); (**D**) *Prevotellaceae;g_*(genus). All data at 7, 9, 13, 21, 28, 32, 42, 51, 58, and 67 weeks old are plotted as weighted and unweighted UniFrac analyses. * *p* < 0.05, *** *p* < 0.001; rice group vs. barley group by unpaired *t*-test.

**Table 1 nutrients-11-01770-t001:** Composition of the standard, rice, and barley diets.

	Standard Diet AIN-93G	Rice Diet	Barley Diet	Mixed Barley Diet (Rice Diet: Barley Diet = 1:4)
Milk casein	200	157.5	130.0	135.5
L-Cysteine	3	3	3	3
Corn starch	397.486	0.0	0.0	0.0
Pregelatinized corn starch	132	0.0	0.0	0.0
Sucrose	100	100	100	100
Soybean oil	70	65.8	57.8	59.4
Lard		50	50	50
Cellulose	50	45.8	0.0	9.2
Pregelatinized rice	-	530.5	-	106.1
Pregelatinized barley	-	-	611.8	489.4
AIN-93 mineral mix	35	35	35	35
AIN-93 vitamin mix	10	10	10	10
Choline bitartrate	2.5	2.5	2.5	2.5
*tert*-butylhydroquinone	0.014	0.014	0.014	0.014

The diet composition is indicted in g/kg diet.

**Table 2 nutrients-11-01770-t002:** Plasma cholesterol levels.

Plasma Cholesterols	Rice Group (mg/dL)	Barley Group (mg/dL)	*p* value
Total LDL-Cho	26.7 ± 2.1	21.1 ± 1.9	0.059
large + medium LDL-Cho	9.4 ± 2.1	5.7 ± 0.5	0.002
small + very small LDL-Cho (high-risk)	17.2 ± 2.1	15.4 ± 1.9	0.522
Total HDL-Cho	114.4 ± 5.6	118.2 ± 4.0	0.582
very large + large HDL-Cho	50.9 ± 4.4	50.0 ± 3.2	0.871
medium + small + very small HDL-Cho (low risk)	63.5 ± 1.7	68.2 ± 1.5	0.045

Plasma cholesterol levels were analyzed by LipoSEARCH^®^ at 16 weeks old. *p*-value: rice group vs. barley group by unpaired *t*-test. LDL-Cho; low-density lipoprotein cholesterol, HDL-Cho; high-density lipoprotein cholesterol.

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
