# Peer review of "Association of Lifelong Intake of Barley Diet with Healthy Aging: Changes in Physical and Cognitive Functions and Intestinal Microbiome in Senescence-Accelerated Mouse-Prone 8 (SAMP8)"

_nutrients, 2019, doi:10.3390/nu11081770_

Round 1
Reviewer 1 Report
General comments
Overall, the paper is well-written and clear. I find the results very interesting and particularly important towards the journal’s mission and our current public health issue of obesity.
Title
-I suggest renamed “Effects” to “Associations”
Abstract
-Well-written
Introduction
-Well-written
Methods
-There is a typo in the website (“QIME.org”) provided for taxonomic classification. I think it should be “QIIME.org”.
Results
-As a general comment, I suggest the authors avoid using the phrases “x affected y” and replace with “x was associated with y”, where “x” could be group or age and y could be an aging phenotype. The current statistical approach does not fully account for causality. Even with significant group by time interactions, there is no account for within-group correlations of time points. However, the interaction results do point to strong temporal associations that highly suggest causality.
Discussion
-I think the authors may consider shifting their suggestions of incorporating barley diet into the public’s lifestyles to rather advocating for more studies in humans. The results suggests that barley-rich diets improve age-related health in mice and provides a strong rationale to replicate these findings across barley types and in regularly-aging animals and human studies.
Figures
-I suggest that the authors add a figure or a supplementary figure of the assessments done throughout the weeks, similar to the figure showing the timing of the feeding.
-For figure 1, it looks like variability is plotted (SD, perhaps), is this correct? Please note what is being presented (e.g., means, SD).
-Though there were no significant differences, I would prefer the authors adding in a Figure 2D showing the spinal curvature data.
-It will benefit the reader if the authors rename the y-axis of Figure 2B to say “Scores of hair glossiness loss”.
Author Response
Dear Reviewer 1,
Thank you for your helpful suggestions. Our point-by-point responses to your comments are provided below.
Our responses to the reviewers are colored blue in responses to reviewer comments. We used the "Track Changes" function in the revised manuscript to show our revised points according to reviewer’s comments, the other changes of wording, and the reduction of total duplicate rate.
Title -I suggest renamed “Effects” to “Associations”
Thank you for your suggestion. We renamed “Effects” to “Associations” in the Title.
Methods -There is a typo in the website (“QIME.org”) provided for taxonomic classification. I think it should be “QIIME.org”.
Thank you for pointing it out. We have to reduce the duplicate rate between this manuscript and our previous report (reference 16). Therefore, we deleted this sentence, and added the following sentence to Section 2.10.: “The procedure was performed as previously reported [16, 32].”
Results
-As a general comment, I suggest the authors avoid using the phrases “x affected y” and replace with “x was associated with y”, where “x” could be group or age and y could be an aging phenotype. The current statistical approach does not fully account for causality. Even with significant group by time interactions, there is no account for within-group correlations of time points. However, the interaction results do point to strong temporal associations that highly suggest causality.
Thank you for your important suggestions. We have now replaced “x affected y” with “x was associated with y” in the results of two-way ANOVA. Moreover, the arrows of the significance of ages in Figures 3A-D, 4, 5A-B, and 6B-G were deleted, because they were misunderstood as within-group correlations. Furthermore, we revised the results of two-way ANOVA obtained from footprint tests.
Discussion
-I think the authors may consider shifting their suggestions of incorporating barley diet into the public’s lifestyles to rather advocating for more studies in humans. The results suggest that barley-rich diets improve age-related health in mice and provides a strong rationale to replicate these findings across barley types and in regularly-aging animals and human studies.
Figures
-I suggest that the authors add a figure or a supplementary figure of the assessments done throughout the weeks, similar to the figure showing the timing of the feeding.
We added green arrows to indicate initiation of mild high-fat diet (at 8 weeks of age) to Figure 1 A–1C, 3A–3D, and 4, and the Figure captions.
-For figure 1, it looks like variability is plotted (SD, perhaps), is this correct? Please note what is being presented (e.g., means, SD).
Thank you for your suggestion. We used SE in Figure 1B. We observed weight loss in some aged SAMP8, and the body weight was quite-variable within the group. Therefore, the SE appeared to be SD.
-Though there were no significant differences, I would prefer the authors adding in a Figure 2D showing the spinal curvature data.
We apologize for the inaccurate numbering of Figure 2. We have corrected the figure number and added a new Figure 3D that shows the spinal curvature data.
-It will benefit the reader if the authors rename the y-axis of Figure 2B to say “Scores of hair glossiness loss”.
Thank you for the suggestion. Thank you for the suggestion. We apologize for the inaccurate numbering of Figure 2. We have now renamed the y-axis of Figure 3B as “Scores of hair glossiness loss”.

Reviewer 2 Report
Glucose tolerance, or behavior test were measured at different life point, why cholesterol was only measured at one time point?
In discussion the authors emphasize the differences in fat absorption with age in HFD paradigms, and in fact offers this explanation for differences in weight among week 60 to 80: Then, it will be also interesting to have the plasma lipid profile
The analysis of microbiome at what time was performed? In methods is not indicated at what time are collected feces.
NORT and OLT: Are the objects in the different time points the same? Familiarization to test, is performed in each time point?
Are the author sure that mice do not have memory form one measure time to other? Data or information about the exploration time in both object have to be informed to be sure that in each point animals are naïve to the objects used.
In the figure 2. How is calculated changes in blood glucose among time points, and which is the relevance? It seems that is showed because there is a significance that can not be showed in the fig 2A
For better understand of age parameters used and its changes table with those parameters have to be added in addition to reference.
The authors mention in several point the presence of diabetic mice. If they were identifies, have the authors considered to mark the results obtained in those mice as outliers?
Is also important to consider that experiment was performed in different experimental condition, is not the same to have several animals in one cage that one isolated animals, indeed when one of the parameters are the influence of diet. Strictly, when mice are housed in the same cage under a dietary intervention, this is a experimental unit (then it can be considered as a n=1 for parameters as weight, food consumption or water consumption), authors indicated that diabetic mice were housed individually…this is another reason to exclude them for the analysis or analyze separately.
Or considered to extract to the group and analyze separately?
In what extend those diabetic mice can bias the final results? This can be discussed.
Discussion
I felt that all the discussion about changes in body weight, are over interpreted, as well as the final recommendation for changing human BMI and age. This is only an animal study that delivers an unusual weight curve that can be explained by several reason 8in fact authors exposed them), but does not enough to change/rethink recommendation at present.
In general, discussion have to be shortened, this part is not a repeat for results but a discussion on the frame of state of the art.
Other: Correct some bibliographic cites
Ie (18): Akiguchi, I.;….
Some works on SAMP8 and glucose intolerance can to be cited
IE
Metabolic stress induces cognitive disturbances and inflammation in aged mice: protective role of resveratrol V Palomera-Ávalos, et al Rejuvenation research 20 (3), 202-217
Two Way ANOVA analysis have to be presentend with data on Freedom degree in ()
Author Response
Dear Reviewer 2,
Thank you for your helpful suggestions. Our point-by-point responses to your comments are provided below.
Our responses to the reviewers are colored blue in this file. We used the "Track Changes" function in the revised manuscript to show our revised points according to reviewer’s comments, the other changes of wording, and the reduction of total duplicate rate.
Glucose tolerance, or behavior test were measured at different life point, why cholesterol was only measured at one time point?
For cholesterol analysis, we sampled approximately 100 µL of blood per mouse from the tail vein. We assumed that multiple samplings might lead to anemia, and affect behavior tests and the lifespan, especially in old age. Although only trace amounts of blood were required for blood glucose measurement using Precision Xceed G3b smart blue electrodes, we measured plasma cholesterols in mice only at the age of 16 weeks (at a young age, to aid recovery from anemia).
In discussion the authors emphasize the differences in fat absorption with age in HFD paradigms, and in fact offers this explanation for differences in weight among week 60 to 80: Then, it will be also interesting to have the plasma lipid profile.
We would like to measure plasma lipid level in old age; however, we could not measure it due to the reasons mentioned in the preceding response. Yamamoto et al. reported the ability of intestinal fat absorption by oral fat tolerance test using the same model SAMP8 mice (Reference 34 in this manuscript). Moreover, they indicated that the length of villus in the small intestine diminishes with age in SAMP8; however, this decrease is inhibited by consumption of a high-fat diet (reference 34).
The analysis of microbiome at what time was performed? In methods is not indicated at what time are collected feces.
Thank you for pointing it out. We added the following sentence of fecal sample collection timings to Section 2.10.: “Fresh fecal samples (approximately 100 mg) were collected at the ages of 7, 9, 13, 21, 28, 32, 42, 51, 58, and 67 weeks”.
NORT and OLT: Are the objects in the different time points the same? Familiarization to test, is performed in each time point?
We used the same objects at different time points, and conducted the same ORT or OLT procedure (habituation, training phase, and test phase) at each time point.
Are the author sure that mice do not have memory form one measure time to other? Data or information about the exploration time in both object have to be informed to be sure that in each point animals are naïve to the objects used.
In our previous study (Reference 15), we determined the suitable objects and test interval period between 1st and 2nd test session of ORT. We confirmed after seven weeks that mice are naïve to the same objects seven weeks ago in the 1st test session of ORT, because there are significant differences between familiar and novel objects in ORT. This finding indicated that the mice forgot the objects used in the previous test, and significantly approached the novel object used seven weeks ago. We added the following sentence to Section 2.9: “The ORT and OLT were performed after more than seven weeks following the previous tests, since the mice forgot the objects used in the previous test, and significantly approached the novel object used seven weeks ago [15].”
In the figure 2. How is calculated changes in blood glucose among time points, and which is the relevance? It seems that is showed because there is a significance that cannot be showed in the fig 2A
There were no significant differences between both groups in Figure 2A. However, the increase in the rice group seemed to be higher than that in the barley group. Therefore, we calculated changes in blood glucose levels from the age of 29 weeks to 42 weeks (age of peak blood glucose level) as shown in Figure 2B.
In Figures 2A and B, not only four diabetic mice, but also the other mice exhibited increased blood sugar with age. Even if the data of severe diabetes were eliminated, the increase in blood sugar were similar (p = 0.007). Therefore, we added the following statement to the Discussion: “The increase in blood sugar was similar level (p = 0.007, data not shown) even without diabetic mice (blood sugar levels > 500 mg/dL)”.
For better understand of age parameters used and its changes table with those parameters have to be added in addition to reference.
Thank you for the suggestion. The reference of aging parameters is in reference 26. As the word numbers of aging parameters affect the increase of duplicate rate of this manuscript, we added the following sentence to the Method: “We evaluated the grading scores as previously reported [26].”, and the reference 26 [26] to the caption of Figure 3 in place of Table. We are sorry about that.
The authors mention in several point the presence of diabetic mice. If they were identifies, have the authors considered to mark the results obtained in those mice as outliers?
Is also important to consider that experiment was performed in different experimental condition, is not the same to have several animals in one cage that one isolated animals, indeed when one of the parameters are the influence of diet. Strictly, when mice are housed in the same cage under a dietary intervention, this is a experimental unit (then it can be considered as a n=1 for parameters as weight, food consumption or water consumption), authors indicated that diabetic mice were housed individually…this is another reason to exclude them for the analysis or analyze separately.
Or considered to extract to the group and analyze separately? In what extend those diabetic mice can bias the final results? This can be discussed.
Thank you for your helpful suggestions. We apologize for the lack of explanation for isolated mice. We isolated the fighting mice at the age of five weeks within one week of beginning the current experiment. The numbers of group-housing (n = 15) and isolated mice (n = 3) in the rice group were equivalent to those in the barley group at the age of five weeks (very young), and were kept until death. We think that the difference in housing size and isolation between both groups were negligible. Therefore, we used all mice data in this experiment.
We corrected the underlined statement “SAMP8 mice with similar body weights were allocated into two groups (rice group: n = 18; barley group: n = 18) and group housed (three mice per cage); however, three fighting mice in each group were isolated at the age of five weeks before being fed with mild high-fat diets”.
SAMP8 mice reportedly suffer from diabetes after being fed with high-fat diets (Reference 36), and with high blood sugar even after being fed a normal diet (new Reference 43). Therefore, we speculated that mice more or less suffered from diabetes before starting this study. The fat energy ratio of high-fat diet is reportedly 50-60%; however, we prepared mild high-fat diets (fat ratio: 27% similar to human diet) for this healthy aging test. We also expected that the degree of diabetes might become milder. We added the following statement in the Discussion: “We speculated that the difference in the rate of severe diabetes indicates prevention of diabetes by barley intake, as three mice in the rice group and one in the barley group suffered from diabetes”.
Discussion
I felt that all the discussion about changes in body weight, are over interpreted, as well as the final recommendation for changing human BMI and age. This is only an animal study that delivers an unusual weight curve that can be explained by several reason 8in fact authors exposed them), but does not enough to change/rethink recommendation at present.
In general, discussion have to be shortened, this part is not a repeat for results but a discussion on the frame of state of the art.
We understand your suggestions. We deleted some sentences in the Discussion. Ms. Amy Xie, Assistant Editor of Nutrients sent me an e-mail of announcement about “Natural and Dietary Agents for Human Diseases Prevention” on 13 June. I informed the title of this manuscript, and received the response from her on 25 June. Therefore, we submitted this manuscript to the special issue “Natural and Dietary Agents for Human Diseases Prevention”. We believe that this animal study contributes to this issue, and should discuss the relation of our results to human study.
The lifelong clinical study is impractical in humans. Therefore, animal studies, as pre-clinical studies, are crucial to understand barley intake in humans. SAMP8 mice are the reported models of dementia or locomotor defects, and are suitable for use as a healthy aging model. Therefore, we examined the temporary changes that occurred during the lifespan rather than a few time points within a certain period to avoid misleading results. Particularly, suitable BMI to age for indicating healthy aging is a critical issue in super-aging societies such as Japan (References 39 and 40). We believe that several aspects of long-term barley intake on healthy human aging can be extrapolated from animal data. We examined many phenotypes, such as behavioral traits associated with age. Therefore, we needed to discuss this points in the Discussion, although we understand what your concerns.
Other: Correct some bibliographic cites
Ie (18): Akiguchi, I.;….
We corrected the name. Thank you for pointing it out.
Some works on SAMP8 and glucose intolerance can to be cited
IE
Metabolic stress induces cognitive disturbances and inflammation in aged mice: protective role of resveratrol V Palomera-Ávalos, et al Rejuvenation research 20 (3), 202-217
Thank you for your suggestion. We added a new Reference 59 based on your suggestion.
Two Way ANOVA analysis have to be presentend with data on Freedom degree in ()
We added freedom degree data to two way ANOVA analysis in the footprint test. Thank you for pointing it out.

Reviewer 3 Report
This is a study aiming to evaluate the effects of barley on lifespan, behavioral phenotype (such as locomotor actvity, balancing power, foot strides), cognitive functions, intestinal microbioma, and metabolic profile in the Senescence-Accelerated Mouse-Prone 8 (SAMP8) mouse. This aim comes from the lack of information on the impact of lifelong barley intake on each of the outcomes reported below.
Authors reported that barley intake in SAMP8 mice significantly improved HDL and LDL-Chol levels, prolonged the lifespan by 4 weeks, and delayed aging-related phenotypes. Thus, its consumption may have positive effects during the healthy aging process.
The study is interesting and well described, but some clarifications are needed:
The nutrient compositon of the diets is reported only in the text. It could be more informative and clear to add a table, also as supplemental materials.
Table 2 is reported in the manuscript but not described in the text. Please add a description in the text.
Author Response
Dear Reviewer 3,
Thank you for your helpful suggestions. Our point-by-point responses to your comments are provided below.
Our responses to the reviewers are colored blue in this file. We used the "Track Changes" function in the revised manuscript to show our revised points according to reviewer’s comments, the other changes of wording, and the reduction of total duplicate rate.
The study is interesting and well described, but some clarifications are needed:
The nutrient composition of the diets is reported only in the text. It could be more informative and clear to add a table, also as supplemental materials.
We added a Supplementary table 1 listing the nutritional content of the diets. In addition, we corrected approximately “25%” of fat energy ratio to “27%”, as shown more precisely in the text.
Supplementary table 1
Rice diet | Barley diet | Rice diet:Barley diet=1:4 | |
Protein | 174 | 174 | 174 |
Fat | 120 | 120 | 120 |
Sugar | 543 | 544 | 544 |
Dietary fiber | 50 | 79 | 73 |
Table 2 is reported in the manuscript but not described in the text. Please add a description in the text.
Thank you for your comment. We apologize for the lack of explanation for Table 2. We added the following sentences in the manuscript (Section 3.2 of the Results): “The total LDL-Cho level in the barley group tended to be lower than that in the rice group (p = 0.059). The “no high-risk LDL-Cho (large + medium)” in the barley group was significantly lower than that in the rice group (p = 0.002), but not the “high-risk LDL-Cho (small + very small)” (p = 0.522). Contrastingly, there were no significant differences in total HDL-Cho levels between both the groups (p = 0.582). The “low risk HDL-Cho” (medium + small + very small particle size) in the barley group was significantly higher than that in the rice group (p = 0.045), but not the “low-risk LDL-Cho (very large and large)” (p = 0.871).”

Round 2
Reviewer 2 Report
The autors had addressed most of my concerns
Author Response
Dear Reviewer 2,
I corrected the spelling error of spelling errors of "Association" to "Association".
Thank you very much for pointing it out.
I attached the revised manuscript.
